

# [1] $^{14}$C observations of atmospheric $CO_2$ at Anmyeondo GAW

# [2] station, Korea: Implications for fossil fuel $CO_2$ and emission

# [3] ratios

[4] Haeyoung Lee[1,2], Edward J. Dlugokencky[3], Jocelyn C Turnbull [4,5], Sepyo Lee[1], Scott J. Lehman[6],
[5] John B Miller[3], Gabrielle Petron[3,5], Jeongsik Lim[7], Gang-Woong Lee[2], Sang-Sam Lee[1] and
[6] Young-San Park[1]
[7]

[8] *Correspondence to Haeyoung Lee (leehy80@korea.kr)*

[9]

[10] [1]National Institute of Meteorological Sciences, Jeju, 63568, Republic of Korea
[11] [2]Atmospheric Chemistry Laboratory, Hankuk University of Foreign Studies, Gyeonggi-do, 17035, Republic of
[12] Korea
[13] [3]NOAA, Earth System Research Laboratory, Global Monitoring Division, Boulder, Colorado, USA
[14] [4] National Isotope Center, GNS Science, Lower Hutt, New Zealand
[15] [5]CIRES, University of Colorado, Boulder, Colorado, USA
[16] [6]INSTAAR, University of Colorado, Boulder, Colorado, USA
[17] [7]Korea Research Institute of Standard and Science, Daejeon, 34113, Republic of Korea
[18]

[19] *Abstract. To understand Korea's carbon dioxide ($CO_2$) emissions and sinks as well as those of*

[20] *the surrounding region, we used 70 flask-air samples collected during May 2014 to August 2016*

[21] *at Anmyeondo (AMY, 36.53° N, 126.32° E; 46 m a.s.l) World Meteorological Organization*

[22] *(WMO) Global Atmosphere Watch (GAW) station, located on the west coast of South Korea, for*

[23] *analysis of observed $^{14}$C in atmospheric $CO_2$ as a tracer of fossil fuel $CO_2$ contribution ($C_{ff}$).*

[24] *Observed $^{14}$C/C ratios in $CO_2$ at AMY varied from -59.5 to 23.1‰ with the measurement*

[25] *uncertainty of ±1.8‰. The derived mean value $C_{ff}$ of (9.7±7.8) μmol mol$^{-1}$ (1σ) is greater than*

[26] *that found in earlier observations from Tae-Ahn Peninsula (TAP, 36.73° N, 126.13° E, 20 m*

[27] *a.s.l., 24 km away from AMY) of (4.4±5.7) μmol mol$^{-1}$ from 2004 to 2010. The enhancement*

[28] *above background of sulfur hexafluoride ($\Delta x(SF_6)$) and carbon monoxide ($\Delta x(CO)$) correlate*

[29] *strongly with $C_{ff}$ (r > 0.7) and appear to be good proxies for fossil fuel $CO_2$ at regional and*



*continental scales. Samples originating from the Asian continent had greater $\Delta x(CO){:}C_{ff}$ ($R_{CO}$)*
*values, (29±8) to (36±2) nmol μmol$^{-1}$, than in Korean local air ((8±2) nmol μmol$^{-1}$). Air masses*
*originating in China showed (1.8±0.2) times greater $R_{CO}$ than a bottom-up inventory suggesting*
*that China's CO emissions are underestimated in the inventory. However, both $R_{CO}$ derived from*
*inventories and observations have decreased relative to previous studies, indicating that*
*combustion efficiency is increasing in both China and South Korea.*
**1 Introduction**
Carbon Dioxide ($CO_2$) is the principle cause of climate change in the industrial era, and is
increasing in the atmosphere at (2.4±0.5) μmol mol$^{-1}$ a$^{-1}$ in a recent decade globally
(www.esrl.noaa.gov/gmd/ccgg/trends/, last access: 6 December 2019). This increase is in fact
predominantly an anthropogenic disturbance that has been demonstrated through $^{14}$C analysis of
tree rings from the last two centuries (Stuiver and Quay, 1981; Suess, 1955; Tans et al., 1979),
caused by accelerated release of $CO_2$ from fossil fuel burning. Atmospheric measurement
program for the ratio $^{14}$C/C in $CO_2$ was initiated in the 1950s and 1960s (Rafter and Fergusson,
1957; Nydal, 1996). Observed $^{14}$C/C ratios are reported in Delta notation ($\Delta(^{14}CO_2)$) as
fractionation-corrected permil (or ‰) deviations from the absolute radiocarbon standard
(Stuiver and Polach, 1977). Many studies show that the variation of $\Delta(^{14}CO_2)$ is an unbiased and
now widely used tracer for $CO_2$ emitted from fossil-fuel combustion (Levin et al., 2003; Turnbull
et al., 2006; Graven et al., 2009; Miller et al., 2012). Therefore measurements of $\Delta(^{14}CO_2)$ are
important to test the effectiveness of emission reduction strategies to mitigate the rapid



atmospheric $CO_2$ increase, since they can partition observed $CO_2$ enhancements, $\Delta x(CO_2)$, into
fossil fuel $CO_2$ ($C_{ff}$) and biological $CO_2$ ($C_{bio}$) components with high confidence (Turnbull et al.,

52   2006).

When trace gases are co-emitted with $C_{ff}$, correlations of their enhancements with $C_{ff}$ improve
understanding of the emission sources of both $C_{ff}$ and the co-emitted tracers. For example, CO
and $CH_4$ emission inventories are typically more uncertain than the fossil fuel $CO_2$ emission
inventory, since those emissions related to complete combustion are generally well estimated
while emissions related to incomplete combustion and agricultural activities are poorly
constrained (Kurokawa et al., 2013). Temporal changes in the observed emission ratio of a trace
gas to $C_{ff}$ can be used to examine emission trends in the trace gas (Tohijima et al., 2014).
Therefore the observed emission ratios of trace gases to $C_{ff}$ can be used to evaluate bottom-up
inventories of various trace gases (e.g., Miller et al., 2012). Here, we used two trace gases,
carbon monoxide (CO) and sulfur hexafluoride ($SF_6$) for this analysis. CO is produced along
with $CO_2$ during incomplete combustion of fossil fuels and biomass. CO enhancements above
background ($\Delta x(CO_2)$) correlate well with $C_{ff}$ and have been used as a fossil fuel tracer
(Gamnitzer et al., 2006; Turnbull et al., 2011a; Turnbull et al., 2011b; Tohijima et al., 2014). $SF_6$
is an entirely anthropogenic gas and is widely used as an arc quencher in high-voltage electrical
equipment (Geller et al., 1997). At regional to continental scales, persistent small leaks to the
atmosphere of $SF_6$ are typically co-located with fossil fuel $CO_2$ sources and allow $SF_6$ to be used
as an indirect $C_{ff}$ tracer, if the leaks are co-located with $C_{ff}$ emissions at the location and scale of
interest (Turnbull et al., 2006; Rivier et al., 2006).
South Korea is a rapidly developing country with fast economic growth, and it is located next to
China, which is the world's largest emitter of anthropogenic $CO_2$, according to the Emissions





Database for Global Atmospheric Research EDGAR (Janssens-Maenhout et al., 2017). The first
$\Delta(^{14}CO_2)$ measurements in South Korea were reported by Turnbull et al. (2011a) based on air
samples collected during October 2004 to March 2010 at Tae-Ahn Peninsula (TAP, 36.73° N,
126.13° E, 20 m a.s.l.). This study showed that observed $CO_2$ at this site was often influenced by
Chinese emissions and the observed ratio of $\Delta x(CO):C_{ff}$ ($R_{CO}$) was greater than expected from
bottom-up inventories. However South Korean $\Delta(^{14}CO_2)$ data are still limited and the ratio of the
other trace gases to $C_{ff}$ barely discussed.
Here we use whole-air samples collected in glass flasks during May 2014 to August 2016 at
Anmyeondo (AMY, 36.53° N, 126.32° E; 46 m a.s.l.) World Meteorological Organization
(WMO) Global Atmosphere Watch (GAW) station, located on the west coast of South Korea and
about 28 km SSE of TAP, where the first study was conducted. We decompose observed $CO_2$
enhancements into their fossil fuel and biological components at AMY to understand sources and
sinks of $CO_2$. We also implemented cluster analysis using the NOAA Hybrid Single Particle
Lagrangian Integrated Trajectory Model (HYSPLIT) to calculate back-trajectories for sample
times and dates. Based on clusters of trajectories from specific regions, trace gas enhancement:
$C_{ff}$ ratios and correlation coefficients were analyzed, especially focused on $SF_6$ and CO, to
determine the potential of alternative proxies to $\Delta(^{14}CO_2)$. Finally we compared our $\Delta x(CO):C_{ff}$
ratio with ratios determined from bottom-up inventories (EDGARv4.3.2) to evaluate reported
CO emissions and how they've changed since 2010.
**2. Materials and Method**
**2.1 Sampling site and methods**



The AMY GAW station is managed by the National Institute of Meteorological Sciences (NIMS)
in the Korea Meteorological Administration (KMA). It has the longest record of continuous $CO_2$
measurement in South Korea, beginning in 1999. It is located on the west coast of Korea about
130 km southwest of the megacity of Seoul, whose population was 9.8 million in 2017.
Semiconductor and other industries exist within a 100 km radius of the station. Also, the largest
thermal power plants fired by coal and heavy oil in South Korea are within 35 km to the
northeast and southeast of the station. The closest town, around 30 km to the east of AMY, is
well known for its livestock industries. Local economic activities are related to agriculture, e.g.,
production of rice paddies, sweet potatoes, and onions, and the area is also known for its leisure
opportunities that increase traffic and tourists in summer, indicating the complexity of
greenhouse gas sources around AMY. On the other hand, air masses often arrive at AMY from
the west and south, which is open to the Yellow Sea. Therefore AMY observes enhanced $CO_2$
compared to many other East Asian stations due not only to numerous local sources but also
long-range transport of air-masses from the Asian continent (Lee et al., 2019).
Two pairs of flask-air samples (4 flasks total, 2 L, borosilicate glass with Teflon O-ring sealed
stopcocks) were collected about weekly from a 40 m tall tower at AMY, regardless of wind
direction and speed from May 2014 to August 2016, generally between 1400 to 1600 local time
(Table S1). A total of 70 sets were collected and analyzed at the National Oceanic and
Atmospheric Administration/Earth System Research Laboratory/Global Monitoring Division
(NOAA/ESRL/GMD) for $CO_2$, CO, and $SF_6$ and for $\Delta(^{14}CO_2)$ by University of Colorado
Boulder, Institute of Arctic and Alpine Research (INSTAAR). NOAA/ESRL/GMD analyzed
$CO_2$ using a non-dispersive infrared analyzer, $SF_6$ using gas chromatography (GC) with electron
capture detection, and CO by vacuum UV, resonance fluorescence. All analyzers were calibrated



with the appropriate WMO mole fraction scales (WMO-X2007 scale for $CO_2$, WMO-X2014A
scale for CO, and WMO-X2014 for $SF_6$; https://www.esrl.noaa.gov/gmd/ccl/, last access: 4
December 2019). The measurement and analysis methods for those gases are described in detail
(http://www.esrl.noaa.gov/gmd/ccgg/behind_the_scenes/measurementlab.html, last access: 4
December 2019). Measurement uncertainties for $CO_2$ and $SF_6$ are reported as 68% confidential
intervals. For $CO_2$, it is 0.07 μmol $mol^{-1}$ for all measurements used here. For $SF_6$, it is 0.04 up to
12 pmol $mol^{-1}$, and undefined above that. For CO, measurement uncertainty has not yet been
formally evaluated, but is estimated at 1 nmol $mol^{-1}$ (68% confidence interval). All $CO_2$, $SF_6$ and
CO data at AMY can be downloaded through ftp://aftp.cmdl.noaa.gov/data/trace_gases/.
The analysis methods for $\Delta(^{14}CO_2)$ are described by Lehman et al.(2013). Measurement
repeatability of $\Delta(^{14}CO_2)$ in aliquots of whole air extracted from surveillance cylinders is 1.8‰
(1 σ), roughly equating to 1 μmol $mol^{-1}$ $C_{ff}$ detection capability from the measurement
uncertainty alone. The $\Delta(^{14}CO_2)$ data at AMY was suggested in Table S1.
**2.2 Data analysis method using $\Delta(^{14}CO_2)$ data**
**2.2.1 Calculation of $C_{ff}$ and $C_{bio}$**
As Turnbull et al. (2009) suggested the observed $CO_2$ ($C_{obs}$) at AMY can be defined as:
$$C_{obs} = C_{bg} + C_{ff} + C_{other} \quad (1)$$
where $C_{bg}$, $C_{ff}$ and $C_{other}$ are the background, recently added fossil fuel $CO_2$ and the $CO_2$ derived
from the other sources.


According to Tans et al. (1993), the product of $CO_2$ abundance and its isotopic ratio is conserved;
the isotopic mass balance can be described as below:
$$\Delta_{obs} C_{obs} = \Delta_{bg} C_{bg} + \Delta_{ff} C_{ff} + \Delta_{other} C_{other} \quad (2)$$
where $\Delta$ is the $\Delta^{14}C$ of each $CO_2$ component of Equ. (1).
Therefore we can calculate fossil fuel $CO_2$ by combining equations (1) and (2) as:
$$C_{ff} = \frac{C_{bg}(\Delta_{obs} - \Delta_{bg})}{\Delta_{ff} - \Delta_{bg}} - \frac{C_{other}(\Delta_{other} - \Delta_{bg})}{\Delta_{ff} - \Delta_{bg}} \quad (3)$$
Fossil fuel derived $CO_2$ contains no $^{14}C$ because the half–life of $^{14}C$ is (5700±30) years (Godwin,
1962) while these fuels are hundreds of millions of years old. As we mentioned in the section 1,
$\Delta(^{14}CO_2)$ is reported as a per mil (‰) deviation from the absolute radiocarbon reference standard
corrected for fractionation and decay with a simplified form; $\Delta(^{14}C)$
$\approx [(^{14}C/C)sample/(^{14}C/C)standard - 1]1000‰$. Therefore $\Delta_{ff}$ is set at -1000‰ (Stuiver and
Pollach, 1977). Background values ($\Delta_{bg}$) in equations (1) to (3) are determined from
measurements from background air collected at Niwot Ridge, Colorado, a high altitude site at a
similar latitude as AMY (NWR, 40.05° N, 105.58° W, 3,526 m a.s.l.). Turnbull et al. (2011a)
showed that the choice of background values did not significantly influence derived
enhancements due to the large regional and local signal at TAP, 28 km from AMY. NWR
$\Delta(^{14}CO_2)$ and other trace gas background values are selected using a flagging system to exclude
polluted samples (Turnbull et al., 2007), and then fitted with a smooth curve following Thoning
et al. (1989).





The second term of equation (3) is typically a small correction for the effect of other sources of
$CO_2$ that have a $\Delta^{14}C$ differing by a small amount that of the atmospheric background, such as
$CO_2$ from 1) nuclear power industry, 2) oceans, 3) photosynthesis and 4) heterotrophic
respiration.
1) The nuclear power industry produces $^{14}C$ that can influence the $C_{ff}$ calculation. South Korea
has nuclear power plants along the east coast that may influence AMY air samples when air-
masses originated from the eastern part of Korea (Figure 1). It is also possible that Chinese
nuclear plants could influence some samples. Here we did not make any correction for this since
most nuclear installations in this region are pressurized water reactors, which produce mainly $^{14}C$
in $CH_4$ rather than $CO_2$ (Graven and Gruber, 2011). 2) For the ocean, although there may also be
a small contribution from oceanic carbon exchange across the Yellow Sea, we consider this
effect small enough to ignore (Turnbull et al., 2011a). Larger scale ocean exchange and also
stratospheric exchange affect both background and observed samples equally, so they can be
ignored in the calculations. 3) For the photosynthetic terms, $^{14}C$ in $CO_2$ accounts for natural
fractionation during uptake, so we also set this observed value the same as the background value.
4) Therefore we only consider heterotrophic respiration. For land regions, where most fossil fuel
emissions occur, heterotrophic respiration could be a main contributor to the second term of
equation (3) due to large $^{14}C$ disequilibrium potentially. When this value is ignored, $C_{ff}$ would be
consistently underestimated (Palstra et al., 2008; Riley et al., 2008; Hsueh et al., 2007; Turnbull
et al., 2006). For this, corrections were estimated as (-0.2±0.1) µmol mol$^{-1}$ during winter and (-
0.5±0.2) µmol mol$^{-1}$ during summer (Turnbull et al., 2009; Turnbull et al., 2006).





$CO_2$ enhancements relative to baseline $CO_2$ are defined as $\Delta x(CO_2)$, with the excess signal of
$C_{obs}$ minus $C_{bg}$ in Equ.(1). Partitioning of $\Delta x(CO_2)$ into $C_{ff}$ and $C_{bio}$ is calculated simply from the
residual of the difference between observed $\Delta x(CO_2)$ and $C_{ff}$.

### 179     2.2.2 The ratio of trace gas enhancement to $C_{ff}$ and its correlation

To obtain the correlation coefficient (r) between $C_{ff}$ and other trace gas enhancements ($\Delta x(x) =$
$x_{obs} - x_{bg}$) and the ratio of any trace gas to $C_{ff}$ ($R_{gas}$), we use reduced major axis (RMA) regression
analysis. The distributions of $R_{gas}$ are normally broad and non-Gaussian and RMA analysis is a
relatively robust method of calculating the slope of two variables that show some causative
relationship. Here, $x_{bg}$ was derived from NWR with the same method described in section 2.2.1.
Therefore, the slope of the linear regression of the RMA fit can be expressed as
$$R_{gas} = \sqrt{\frac{\sum \Delta x(x)^2 - (\sum \Delta x(x))^2/n}{\sum C_{ff}^2 - (\sum C_{ff})^2/n}} \quad (4)$$

And the uncertainty of $R_{gas}$ is defined as
$$U = \sqrt{\frac{\sum (\Delta x(x) - \Delta x(x)\prime)^2/n}{\sum C_{ff}^2 - (\sum C_{ff})^2/n}} \quad (5)$$
Here, $\Delta x(x)\prime = R_{gas} \times (C_{ff} - \overline{C_{ff}}) + \overline{\Delta x(x)}$

The correlation coefficient is expressed,


$$r = \sqrt{\frac{(\sum \Delta x(\mathrm{x}) C_{\mathrm{ff}} - \frac{\sum \Delta x(\mathrm{x}) \sum C_{\mathrm{ff}}}{n})^2}{(\sum \Delta x(\mathrm{x})^2 - \frac{(\sum \Delta x(\mathrm{x}))^2}{n}) \times (\sum C_{\mathrm{ff}}^2 - \frac{(\sum C_{\mathrm{ff}})^2}{n})}} \quad (6)$$
Results for each species are given in Table 1.
**2.3 HYSPLIT cluster analysis**
HYSPLIT trajectories were run using Unified Model-Global Data Assimilation and Prediction
System (UM-GDAPS) weather data at 25 km by 25 km horizontal resolution to determine the
regions that influence air mass transport to AMY. A total of 70 air-parcel back-trajectories were
calculated for 72-h periods at 3-h intervals matching the time of each flask-air sample taken at
AMY from May 2014 to August 2016. We assign the sampling altitude as 500 m, since it was
demonstrated that HYSPLIT and other particle dispersion back-trajectory models (e.g.,
FLEXPART) are consistent at 500 m altitude (Li et al., 2014). Cluster analysis of the resulting
70 back-trajectories categorized six pathways through which air parcels arrive at AMY during
the time period of interest.
Among the calculated back-trajectories, 67% indicate air masses originating from the Asian
continent. Back-trajectories of continental background air (CB) originating in Russia and
Mongolia occurred 13% of the time. 23% of the trajectories originated and travelled through
northeast China (CN). The CN region includes Inner Mongolia and Liaoning, one of the most
populated regions in China with 43.9 million people in 2012. These CN air masses arrive in
South Korea after crossing through western North Korea. 17% of the trajectories are derived
from central eastern China around the Shandong area (CE). The CE region contains Shandianzi
(SDZ, 40.65° N, 117.12° E, 287 m a.s.l.) located next to the megacities of Beijing and Tianjin,
which are some of China's highest $CO_2$ emitting regions (Gregg et al., 2008). 14% are Ocean





Background (OB) derived from the East China Sea, which passed over the eastern part of China
such as Shanghai. Flow from South Korea also travels through heavily industrialized and/or
metropolitan regions in South Korea (Korea Local, KL, 19%) and under stagnant conditions
(Polluted Local region, PL, 14%). Some of the KL air-masses have also passed over the East Sea
and Japan.
**3. Results and discussions**
**3.1 Observed $\Delta(^{14}CO_2)$ and portioning of $CO_2$ into $C_{ff}$ and $C_{bio}$**
AMY $\Delta(^{14}CO_2)$ values are almost always lower than those observed at NWR, which we consider
to be broadly representative of background values for the mid-latitude Northern Hemisphere
(Figure 2). NWR $\Delta(^{14}CO_2)$ , which is based on weekly air samples, was in the range 10.0 to 21.2
‰, with an average (16.6±3)‰ (1σ) from May 2014 to August 2016. Waliguan (WLG, 36.28°
N, 100.9° E, 3816 m a.s.l.), an Asian background GAW station in China, also showed similar
$\Delta(^{14}CO_2)$ levels to NWR with an average of $(17.1 \pm 6.8)$ ‰ in 2015 (Niu et al., 2016,
measurement uncertainty ±3‰). $\Delta(^{14}CO_2)$ at AMY varied from -59.5 to 23.1‰ and had a mean
value of (-6.2±18.8)‰ (1σ) during the experiment period (Table S1). This was similar to results
from observations at SDZ, which is located about 100 km northeast of Beijing, in the range of -
53.0 to 32.6‰ with an average (-6.8±21.1)‰ during Sep 2014 to Dec 2015 (Niu et al., 2016).





Calculated $C_{ff}$ at AMY ranges between -0.05 and 32.7 µmol mol$^{-1}$ with an average of (9.7±7.8)
µmol mol$^{-1}$ (1σ); high $C_{ff}$ was observed regardless of season (Figure 2 (a)). One negative $C_{ff}$
value of -0.05 µmol mol$^{-1}$ was estimated due to greater AMY $\Delta(^{14}CO_2)$ than NWR on July 30,
2014. Although negative $C_{ff}$ values are non-physical, this value is not significantly different from
zero, and is reasonable given that this air originated from the OB sector. The range of $C_{ff}$ in the
AMY samples is similar to that observed at TAP from 2004 to 2010 (-1.6 to 42.9 µmol mol$^{-1}$
$C_{ff}$), but $C_{ff}$ is on average about twice as high at AMY as in the 2004 to 2010 TAP samples
(mean (4.4±5.7) µmol mol$^{-1}$) (Turnbull et al., 2011a). A more detailed comparison of results
based on differences between samples derived from the Asian continent and Korea local air is
provided in section 3.2.
Estimated $C_{bio}$, as defined in section 2.2.1, varied from -18.1 to 15.7 µmol mol$^{-1}$ (mean (0.9±5.8)
µmol mol$^{-1}$) at AMY (Figure 2 (c)). $C_{bio}$ showed a strong seasonal cycle with the lowest values
from July to September when photosynthetic drawdown is expected to be strongest, in good
agreement with the previous TAP study (Turnbull et al., 2011a). Even though $C_{bio}$ was at times
negative, mainly due to photosynthesis during summer, the largest positive $C_{bio}$ was also
observed in summer.
The largest $C_{ff}$ by season was observed in order of winter (DJF, (11.3±7.6), n=14) > summer
(JJA, (10.7±9.2), n=11) > spring (MAM, (8.6±8.0), n=22) > autumn (SON, (7.6±5.6), n=17) with
a unit of µmol mol$^{-1}$. When we consider only positive contributions of $C_{bio}$ samples, the order
was summer ((4.6±4.0), n=14) > autumn ((4.1±2.5), n=9) > spring ((3.8±2.6), n=13) > winter
((3.4±2.5), n=11) with a unit of µmol mol$^{-1}$.





$C_{ff}$ in summer was nearly as high as in winter. This is because lower wind speeds are observed at
AMY during summer (Lee et al., 2019), suggesting that these summer high values may reflect
emission from local activities, which were described in section 2.1, more than in other seasons.
The highest $C_{bio}$ value was also observed in summer. PL sector showed that positive $C_{bio}$
correlates with $CH_4$, which is a tracer for agriculture when observed in TAP local air masses.
Turnbull et al.(2011a) also showed similar results.
In winter, $C_{bio}$ was relatively lower than in other seasons while $C_{ff}$ was highest. During winter,
AMY is mainly affected by long-range transport of air-masses from China due to the Siberian
high (Lee et al., 2019). Therefore air samples were less affected by local activities in winter but
$C_{bio}$ still contributed almost 23% to $\Delta x(CO_2)$. In the dry season (from October to March), forest
fires, which contribute the largest portion of total $CO_2$ emissions from open fires at the national
scale, are concentrated in northeastern and southern China (Yin et al., 2019). The highest CO
was observed in winter ((449.1±244.1) nmol mol$^{-1}$ (1σ) in winter while (236.8±124.4) nmol
mol$^{-1}$ (1σ) in summer), which also supports biomass burning and bio fuels as large contributors
to observed $CO_2$ enhancements in winter. Turnbull et al. (2011a) also showed that 20-30% of
winter $CO_2$ enhancements at TAP were likely contributed by biofuel combustion, along with
plant, soil, human, and animal respiration.
Regardless of the source, we find that $C_{bio}$ contributes substantially to atmospheric $CO_2$
enhancements at AMY in air masses affected by local and long-range transport, so $CO_2$
enhancements above background cannot be reliably interpreted as entirely due to $C_{ff}$.
**3.2 $C_{ff}$ comparison between Korea Local and Asian Continent samples**



To more clearly identify samples originating from the Asian continent (trajectory clusters CB,
CN, CE, and OB) and Korea Local (trajectory cluster KL) after cluster analysis of the 70 sets of
measurements, we use wind speed data from the Automatic Weather System (AWS) installed at
the same level as the air sample inlet at AMY. Among the data from CB, CN, CE, OB, and KL,
when wind speed was less than 3 m/s, we assumed that those samples could be affected by local
pollution. PL was also ruled out since it was affected by local pollutions under the stagnant
condition. Therefore we use only 41 sets of observations for this analysis (Table 1).
$C_{ff}$ is highest in the order CE > CN > KL > CB > OB (Table 1). During the experimental period,
the averages from Asian continent (sectors CE and CN) were higher than KL without the
baseline level. The calculated mean $C_{ff}$ using only CE, CN, CB and OB, which sample
substantial outflow from the Asian Continent, was (7.6±3.9) µmol mol$^{-1}$.
When we compared the KL samples ((8.6±5.3) µmol mol$^{-1}$) with those from Korea Local air-
masses observed at TAP ((8.5±8.6) µmol mol$^{-1}$, Turnbull et al., 2011a), mean $C_{ff}$ was quite
similar (Figure 3). However, when comparing the $C_{ff}$ values from CB air masses in this study
and TAP far-field (from China) samples (Turnbull et al., 2011a), $C_{ff}$ almost doubled from (2.6±
2.4) to (4.3±2.1) µmol mol$^{-1}$, even though they might be expected to have had similar air mass
back-trajectories. We also compared the values at SDZ from 2009 to 2010 (Turnbull et al., 2011a)
and in 2015 (Niu et al., 2016); they also increased, not only in the samples that were affected by
Beijing and North China Plain (SDZ-BN), which are comparably polluted, but also in the
samples that were affected by northeast China (SDZ-NE). For SDZ-BN samples, $C_{ff}$ increased



from (10±1) to (16±7.6) µmol mol$^{-1}$ from 2009/2010 to 2015. The AMY samples from CE,
which flow over Beijing, showed (11.2±8.3) µmol mol$^{-1}$ of $C_{ff}$ and were also slightly greater
than the 2009 – 2010 SDZ-BN samples (Turnbull et al., 2011a). For SDZ-NE samples, $C_{ff}$ was
(3±7) µmol mol$^{-1}$ in 2009 to 2010 and increased to (7.6±6.8) µmol mol$^{-1}$ in 2015. Since the
SDZ-NE samples are affected by northeast China according to Turnbull et al. (2011a) and Niu et
al. (2016), we also see CN originated from northeast China and it was around (10.6±6.9) µmol
mol$^{-1}$.
It has been suggested that inter-annual variability in observed mean $C_{ff}$ in South Korea could
reflect changing fossil fuel $CO_2$ emissions, or could indicate inter-annual variability in the air
mass trajectories of the (small) dataset of flask-air samples (Turnbull et al., 2011a). Even though
the growth rate of $C_{ff}$ emission has been decreasing slowly in East Asia since 2010 due to
emission reduction policies (Labzovskii et al., 2019), reported emissions increased 16.7% in
China and 1.8% in South Korea from 2010 to 2016 (Janssens-Maenhout et al., 2017). This is
broadly consistent with the flat trend in observed $C_{ff}$ in Korea Local air masses, and in the
upward trend in $C_{ff}$ observed in air-masses flowing out from Asia. Therefore it is possible that
AMY mean $C_{ff}$ increased relative to the earlier TAP observations due to increased fossil fuel
emissions from the Asian continent. It is also likely that the proximity of local emission sources
to AMY is causing higher observed $C_{ff}$ under some synoptic conditions.
On the other hand, those values from this study showed large variability with small sample
numbers, further study will be necessary.





**3.3. Correlation of $C_{ff}$ with $SF_6$ and CO, and their emission ratios**
We calculated correlation coefficients (r from Equ. (6)) between $SF_6$ and CO enhancements with
$C_{ff}$ and their ratios from Equ. (4) with the 50 samples that were described in section 3.2 including
PL sector (n=9) and whose values are tabulated in Table 1.
The correlations of CO enhancements ($\Delta x(CO)$) with $C_{ff}$ were strong (r > 0.7) in all sectors
except PL, while $SF_6$ enhancements ($\Delta x(SF_6)$) correlated strongly with $C_{ff}$ (r > 0.8) for CE and
OB in outflow from the Asian Continent and KL. $R_{CO}$ and $R_{SF6}$ were different between Korea
Local and outflows from the Asian Continent.
For $SF_6$, observed mean levels were high in order of KL > PL > CN and CE > OB > CB (Table
1). $SF_6$ in KL and PL were higher than from the Asian Continent, since South Korea has larger
$SF_6$ emissions than most countries (ranked at 4[th] as of 2010 according to the EDGAR4.2.)
because of liquid-crystal display (LCD) and electrical equipment production (Fang et al., 2014).
Even though South Korea showed higher $SF_6$, the correlation is different between KL and PL.
Under stagnant conditions, emitted $SF_6$ is less diluted by mixing, so that in PL, $\Delta x(SF_6)$
correlated weakly with $C_{ff}$. On the other hand, KL, CE and OB showed strong correlations (r >
0.8). Those three sectors are also larger $SF_6$ sources compared to other regions, according to $SF_6$
emission estimates for Asia (Fang et al., 2014). Because long-range transport allows time for
mixing, $SF_6$ and $C_{ff}$ emissions are effectively co-located at not only continental scales but also
regional scales. Thus $SF_6$ can be a good tracer of fossil fuel $CO_2$.
Even though the correlation between $\Delta x(SF_6)$ and $C_{ff}$ was strong in CE, OB and KL, $R_{SF6}$ is
different between South Korea and outflow from the Asian continent (Figure S2). In a previous
study, observed $R_{SF6}$ was 0.02 to 0.03 pmol µmol$^{-1}$ at NWR in 2004 (Turnbull et al., 2006). Here,



the ratio was at (0.19±0.03) and (0.17±0.03) pmol µmol$^{-1}$ for CE and OB respectively. For KL,
it was (0.66±0.16) pmol µmol$^{-1}$ indicating much larger ratios than in outflow from the Asian
continent. Further, observed $R_{SF6}$ is 2 to 3 times greater for all air masses than predicted from
bottom-up inventories based on national scale roughly. For this calculation, we use EDGAR4.3.2
for $CO_2$ and EDGAR4.2 for $SF_6$. We repeat the calculations for both $CO_2$ and $SF_6$ with Korea's
National Inventory Report (KNIR, Greenhouse Gas Inventory and Research Center, 2018).
Using $SF_6$ for 2010 from EDGAR4.2, we obtain $R_{SF6}$ of 0.08 pmol µmol$^{-1}$ for China while for
South Korea it was 0.14 pmol µmol$^{-1}$. Especially for South Korea, this is much lower than the
observed $R_{SF6}$. When KL $R_{SF6}$ was compared to ratios calculated from the KNIR inventory (0.27
pmol µmol$^{-1}$ for 2010 and 0.22 pmol µmol$^{-1}$ for 2014), it was closer to observed $R_{SF6}$ than
EDGAR, but still underestimated (Figure S3 and S2). This result suggests that the observed ratio
could be used to re-evaluate the bottom-up inventories (Rivier et al., 2006), especially targeting
the Asian continent. Since KL $R_{SF6}$ showed greater uncertainty than CE and OB, it would be
useful to get more data to try and derive a more robust estimate to evaluate $SF_6$ emissions in
Korea.
High CO was mainly observed in outflow from the Asian continent in order of CE > CN > PL >
CB > KL > OB (Table 1). The order of CO is quite different to that of $SF_6$. CO from KL and PL
is lower than from outflow from the Asian continent, except for the OB sector, indicating that
high CO can be a tracer of outflow from the Asian continent. Since CO is produced during
incomplete combustion, it is more closely related to fossil fuel $CO_2$ emissions than the other
trace gases. Therefore in most cases the correlation between CO and $C_{ff}$ was strong. $R_{CO}$ was
very different between air masses originating from South Korea Local ((8±2) nmol µmol$^{-1}$) and





the Asian continent ((29±8) to (36±2) nmol μmol$^{-1}$), likely due to differences in combustion
efficiencies. The higher continental emission ratios may also result from some contribution of
biofuel combustion and agricultural burning in the Asian continent, which have significantly
higher CO emission than fossil-fuel combustion (Akagi et al., 2011).
Typically CO shows seasonal variations with lower values in summer due to the photochemical
sink. Among the samples, the samples collected in summer were mainly rejected through wind
speed cut-off (less than 3 m/s) since AMY has lower wind speed in summer (Lee et al., 2019).
Only OB sector includes 4 summer samples (of 7), because summer air masses are mainly from
the southern part of the Yellow Sea (Lee et al., 2019). However, we assumed $R_{CO}$ is less affected
by the summer sink, since only two $\Delta x(CO)$ samples were negative for OB (Figure S2) and $R_{CO}$
was consistent whether or not the negative $\Delta x(CO)$ values were considered. We compare our $R_{CO}$
to results from previous studies in section 3.4.
**3.4 Comparison of measured emission ratios to CO inventory data**
To compare emission ratios derived from atmospheric observations with those from inventories
for 2000 to 2012, we calculated inventory emission ratio ($I_{CO/CO2}$) as:
$I_{CO/CO2} = E_{CO}/E_{CO2} \times M_{CO2}/M_{CO}$ (7)
Where, $E_{CO}$ and $E_{CO2}$ are total CO and fossil fuel $CO_2$ emissions in gigagrams (Gg, $10^9$ g) from
the bottom-up national inventory. $M_X$ are the molar masses of CO and $CO_2$ in g mol$^{-1}$.
We use EDGAR4.3.2 (Janssens-Maenhout et al., 2017) and KNIR (Greenhouse Gas Inventory
and Research Center, 2018) for inventory information for both CO and $CO_2$.



The uncertainty of EDGAR4.3.2 emissions was reported as a 95% confidence interval (Janssens-
Maenhout et al., 2019), ±5.4% for China and ±3.6% for South Korea (personal communication
with Dr. Efisio Solazzo). The uncertainties of CO and $SF_6$ emissions were not reported by
EDGAR. For KNIR, the $CO_2$ 2016 emission uncertainty in the energy sector was ±3%
(Greenhouse Gas Inventory and Research Center, 2018). KNIR does not provide uncertainties
for other emission sectors of $CO_2$, nor from emissions of CO and $SF_6$.
In Fig. 4 we confirm that the CO to $C_{ff}$ emission ratios ($R_{CO}$) derived from both observations and
inventories for China and South Korea are decreasing. Since $C_{ff}$ emissions appear to be flat
(South Korea) or slightly increasing (China), this indicates that combustion efficiency and/or
scrubbing of CO is improving.
For South Korea, EDGAR4.3.2 indicated that CO emissions from the energy sector (98% to 99%
of total emission) decreased by 47% between the 1997 and 2012. South Korean fossil fuel $CO_2$
emissions increased until 2011 and remained mostly constant from 2011 to 2016
((603,901±4,315) Gg $CO_2$) (Figure S4). Therefore the decreased trend in the emission ratio
seems to reflect recent decreases in CO emissions in South Korea. Turnbull et al. (2011a)
determined an observed mean $R_{CO}$ of (13±3) nmol µmol$^{-1}$ during 2004 to 2010. Suntharalingam
et al. (2004) estimated $R_{CO}$ 15.4 nmol µmol$^{-1}$ for South Korea in 2001 from $CO_2$ and CO airborne
observations ($C_{ff}$ was not determined). Recently, the KORUS-AQ campaign, which was
conducted over Seoul from May to June in 2016, estimated $R_{CO}$ as 9 nmol µmol$^{-1}$ (Tang et al.,
2018) based on $CO_2$ and CO observations ($C_{ff}$ was not determined). Our study gives $R_{CO}$ of (8±2)





nmol µmol$^{-1}$ for South Korea, slightly but not significantly lower than the KORUS-AQ result for
Seoul. This difference could be due to different source regions (Seoul vs the larger Korean
region) and different experimental periods (two months vs two years). Different contributions of
$C_{bio}$ and $C_{ff}$ to total $CO_2$ may bias the $R_{CO}$ calculation when total $CO_2$ was used in the KORUS-
AQ study (e.g., Miller et al., 2012). The South Korean national $R_{CO}$ from EDGAR4.3.2 in 2012
was 6.7 nmol µmol$^{-1}$, consistent with our observations. Using KNIR for 2016, we obtain $R_{CO}$ of
2.1 nmol µmol$^{-1}$. KNIR seems to have uncounted CO emissions, since it is unreasonably low
during all comparison periods (Figure S5). For example, CO emissions recently derived from
fugitive emissions and residential/other sectors increased to 14% and 11.5% of total emission
respectively in EDGAR but were not reported in KNIR.
For China the inventories estimate that CO emissions from the energy sector, (96.5±0.2)%, were
almost constant through the 1990s, and then increased during the early-2000s from industrial
processes (8.8% of total emissions in 2012). Fossil fuel $CO_2$ emission in China also increased
until 2013 and then stayed roughly constant at (10,461,890±60,571) Gg according to
EDGAR4.3.2. Thus even though both emissions show an increase from 2000 to 2016 for fossil
fuel $CO_2$ and to 2012 for CO, the emission ratio decreased (Figure S4 and Figure 4) seeming to
indicate that combustion efficiency is improving. Many studies observed decreasing $R_{CO}$ in
China from 2000 to 2010 (Turnbull et al., 2011a; Wang et al., 2010). Suntharalingam et al. (2004)
reported $R_{CO}$ was 55 nmol µmol$^{-1}$ in 2001 ($C_{ff}$ was not determined). In the Beijing region, $R_{CO}$
decreased from 57.80 to 37.59 nmol µmol$^{-1}$ during 2004 to 2008 (Wang et al., 2010). The overall
$R_{CO}$ was (47±2) nmol µmol$^{-1}$ at SDZ for 2009-2010 and (44±3) nmol µmol$^{-1}$ in air-masses that
originated from the Asian continent from 2005 to 2009 (Turnbull et al., 2011a). Tohjima et al.


(2014) explained that surface based $R_{CO}$ decreased from 45 to 30 nmol µmol$^{-1}$ in outflow air
masses from China from 1998 to 2010. Fu et al. (2015) also observed $R_{CO}$ of 29 nmol µmol$^{-1}$
over mainland China in 2009. In Beijing, which is located along the path of CE, it was (30.4±1.6)
nmol µmol$^{-1}$ and (29.6±3.2) nmol µmol$^{-1}$ for Xiamen in 2016, which is in the OB sector (Niu et
al., 2018). During KORUS-AQ in 2016, $R_{CO}$ of 28 nmol µmol$^{-1}$ was observed over the Yellow
Sea. Some of those studies did not differentiate $C_{ff}$ from the total $CO_2$ enhancement, so, although
$R_{CO}$ still includes uncertainties, it is continually decreasing.
In this study $R_{CO}$ is (29±8), (31±8), (36±2), and (31±4) nmol µmol$^{-1}$ for CB, CN, CE and OB,
consistent with Tang et al.(2018) and Liu et al.(2018). On the other hand, $R_{CO}$ in CE is higher
than in other sectors in this study. The Shandong area, which is located in the path of CE, has
been plagued with problems of combustion inefficiency and ranked as the largest consumer of
fossil fuels in all of China (Chen and Li, 2009). The uncertainties in our observed $R_{CO}$ for this
region overlap with other sectors such as CB, CN and OB, so further monitoring of the ratios
will help to get more detailed information.
In South Korea and China, atmosphere-based $R_{CO}$ values are 1.2 times and (1.8±0.2) times
greater than in the inventory, respectively. This is in agreement with previous studies (Turnbull
et al., 2011a; Kurokawa et al., 2013; Tohjima et al., 2014). One explanation is that EDGAR does
not reflect secondary CO production, which can be a significant contributor to CO (Kurokawa et
al., 2013). Also, CO derived from biomass burning and biofuels was not included in this
inventory. Therefore, this indicates that top-down observations are necessary to evaluate and
improve bottom-up emission products.



## 4. Summary and conclusion

Observed $\Delta(^{14}CO_2)$ values at AMY ranged from -59.5 to 23.1‰ (a mean value of (-6.2±18.8)

‰ (1σ)) during the study period, almost always lower than those observed at NWR, which we

consider to be broadly representative of background values for the mid-latitude Northern

Hemisphere. This reflects the strong imprint of fossil fuel-$CO_2$ emissions recorded in AMY air

samples. Calculated $C_{ff}$ using $\Delta(^{14}CO_2)$ at AMY ranges between -0.05 and 32.7 µmol mol$^{-1}$ with

an average of (9.7±7.8) µmol mol$^{-1}$ (1σ); this average is twice as high as in the 2004 to 2010

TAP samples (mean (4.4±5.7) µmol mol$^{-1}$) (Turnbull et al., 2011a). We also observed high $C_{ff}$

regardless of the season or source region. After separately identifying samples originating from

the Asian continent and the Korean peninsula, we determined that the mean $C_{ff}$ increased relative

to the earlier observations due to increased fossil fuel emissions from the Asian continent. Note,

however, that our data span a relatively limited time period, so a longer time-series would

increase confidence in tracking this change.

Because $\Delta x(CO)$ and $\Delta x(SF_6)$ agreed well with $C_{ff}$, but showed different slopes for Korea and the

Asian continent, those $R_{gas}$ values can be indicators of air mass origin and those gases can be

proxies for $C_{ff}$. Overall, we have confirmed that both $R_{CO}$ derived from inventory and

observation have decreased relative to previous studies, indicating that combustion efficiency is

increasing in both China and South Korea. Atmosphere-based $R_{CO}$ values are 1.2 times and

(1.8±0.2) times greater than in inventory values for South Korea and China, respectively. This

discrepancy may arise from several sources including the absence of atmospheric chemical CO





production such as oxidation of $CH_4$ and non-methane VOCs. Therefore those values can be
used for improving bottom-up inventory in the future. Finally, we stress that because $C_{bio}$
contributes substantially to $\Delta x(CO_2)$, even in winter, $\Delta^{14}C$-based $C_{ff}$ (and not $\Delta x(CO_2)$) is
required for accurate calculation of both $R_{CO}$ and $R_{SF6}$.



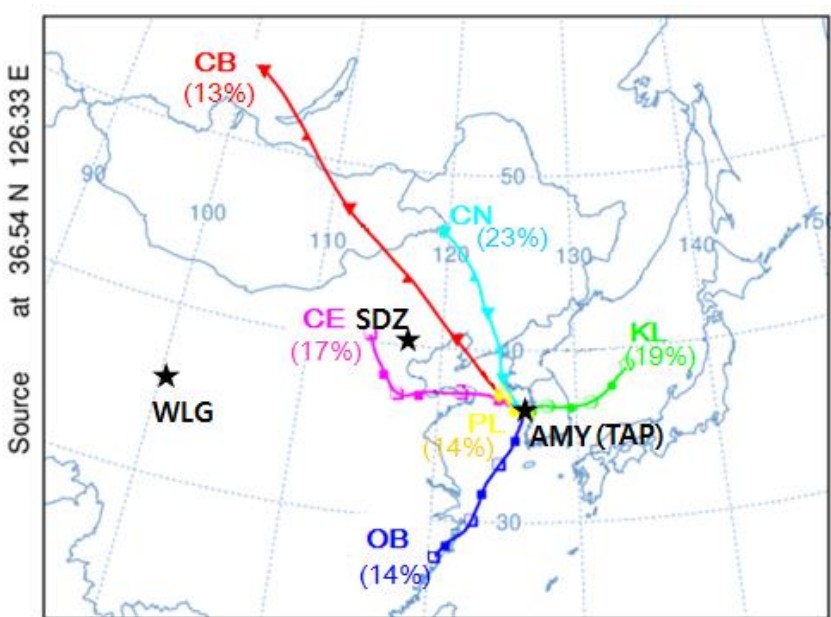


Figure 1. A total of 70 air-parcel back-trajectories were calculated for 72-h periods at 3-h intervals from May 2014 to August 2016 using the HYSPLIT model in conjunction with KMA UM GDAPS data at 25 km by 25 km resolution. Station locations are: WLG (Waliguan, 36.28° N, 100.9° E, 3816 m a.s.l.), SDZ (Shandianzi, 40.65° N, 117.12° E, 287 m a.s.l.), and AMY (Anmyeondo, 36.53° N, 126.32° E, 86 m a.s.l.). TAP (Tae-Ahn Peninsula, 36.73° N, 126.13° E, 20 m a.s.l.) is around 28 km northeast from AMY.






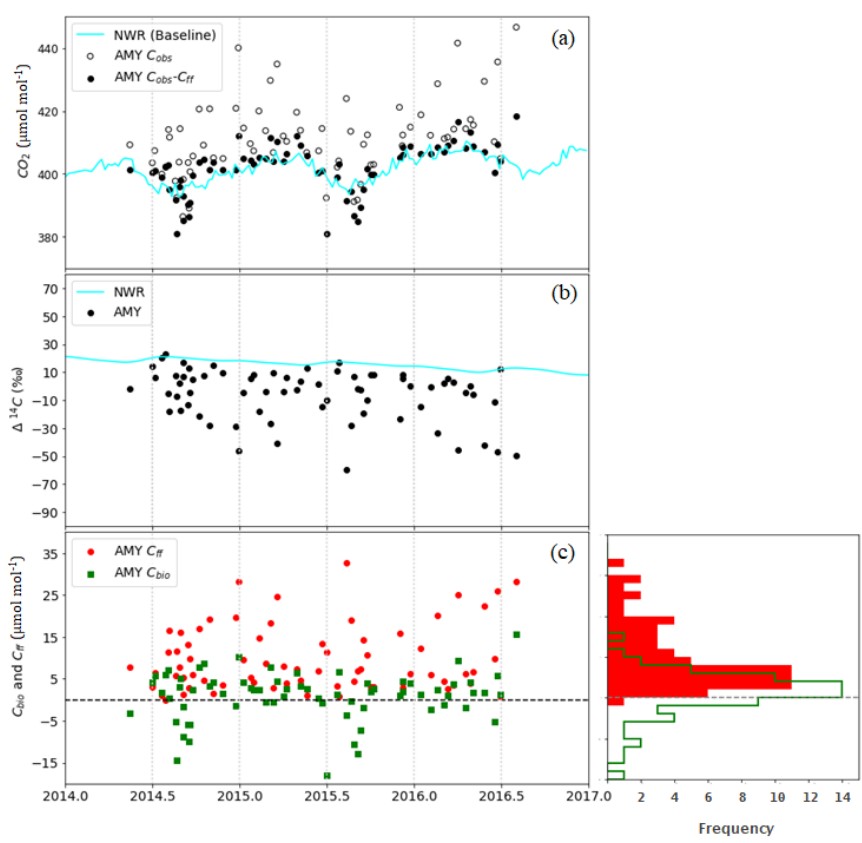

Figure 2. Time series of (a) observed $CO_2$ dry air mole fraction (open circles) and observed $CO_2$
($C_{obs}$) minus $C_{ff}$ calculated from $\Delta(^{14}CO_2)$ (closed circles). (b) $\Delta(^{14}CO_2)$ at AMY (black circles)
and at NWR (Niwot Ridge, line), baseline data. (c) Time series of $C_{ff}$ and $C_{bio}$ calculated from
$\Delta(^{14}CO_2)$ at AMY.



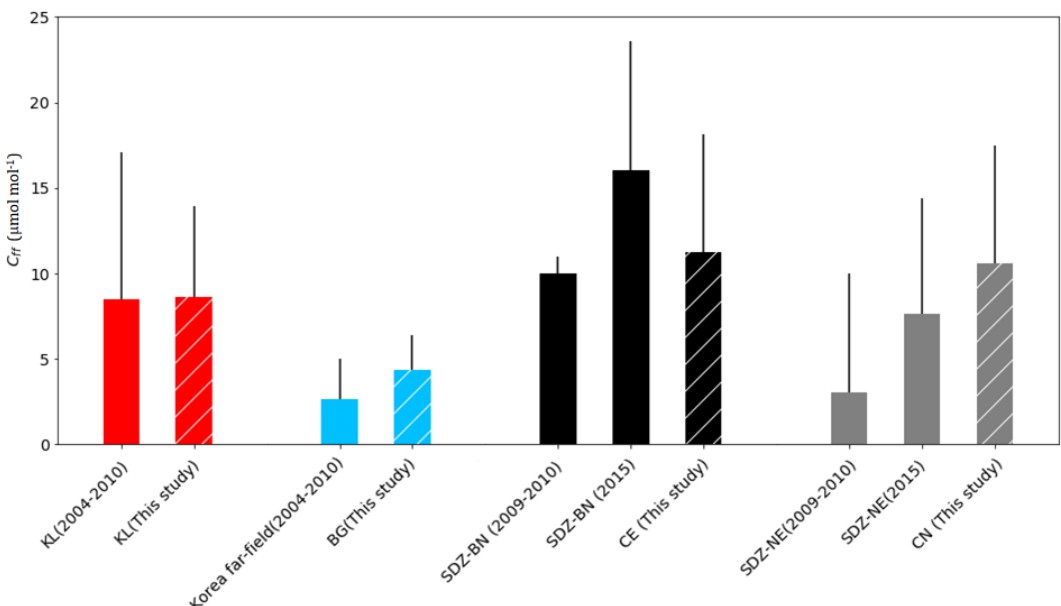


Figure 3. Calculated $C_{ff}$ (µmol mol$^{-1}$). Red bars are for KL and blue bars are for Korea far-field

(China) (2004-2010 from Turnbull et al. (2011a)). Black bars are for SDZ-BN samples that were

affected by Beijing and North China plain. Gray bars for SDZ-NE indicate samples that were

affected by regions northeast of SDZ. SDZ (2009-2010) is from Turnbull et al. (2011a) and SDZ

(2015) is from Niu et al. (2016). Hatched red, blue, black and grey bars are derived from this

study during 2014 to 2016.

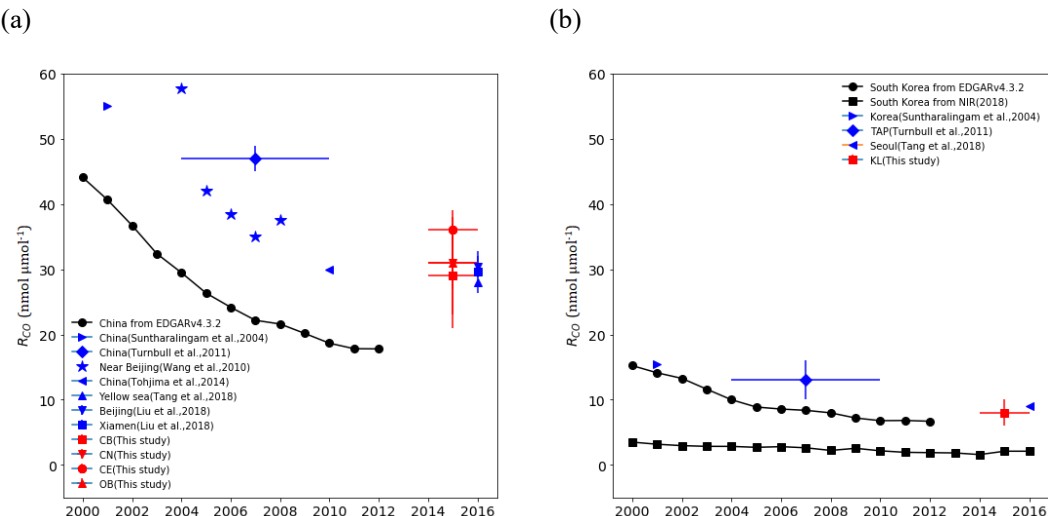

Figure 4. $R_{CO}$ for China (a) and for South Korea (b). Black circles: EDGARv.4.3.2 emission
inventory. Black squares: National Inventory Report, Korea (2018). Blue symbols are from other
studies (Suntharalingam et al., 2004; Wang et al., 2010; Turnbull et al., 2011a; Tohjima et al.,
2014; Liu et al., 2018; Tang et al., 2018). Red symbols: This study. Y-error bars: uncertainty
according to equation (5). X-error bars: the period for the mean value.



Table 1. Means and standard deviations of $C_{ff}$ (μmol mol$^{-1}$), CO (nmol mol$^{-1}$) and SF$_6$ (pmol mol$^{-1}$) (total N=50, without PL N=41). The correlations (r) and the ratio ($R_{gas}$) of enhancement between $C_{ff}$ were determined by Reduced Major Axis (RMA) regression analysis on each scatter plot to obtain regression slopes. The uncertainty of $R_{gas}$ refers to equation (5). When r is less than 0.7, $R_{gas}$ was not included here. N is the number of data. The unit of $R_{CO}$ is nmol μmol$^{-1}$ and for $R_{SF6}$ it is pmol μmol$^{-1}$. A plot of $R_{CO}$ and $R_{SF6}$ is shown in Figure S1.

|  | Outflow from the Asia continent | | | | South Korea | |
|---|---|---|---|---|---|---|
|  | CB (N=7) | CN (N =9) | CE (N =9) | OB (N =7) | KL (N =9) | PL (N =9) |
| $C_{ff}$ | 4.3±2.1 | 10.6±6.9 | 11.2±8.3 | 4.1±2.7 | 8.6±5.3 | 15.6±11.6 |
| CO | 233±59 | 353±219 | 473±293 | 169±90 | 228±40 | 259±100 |
| SF$_6$ | 9.0±0.4 | 10.1±1.2 | 10.1±1.5 | 9.2±0.5 | 13.0±3.3 | 12.7±6.2 |
| $R_{CO}$ (r) | 29±8 (0.80) | 31±8 (0.76) | 36±2 (0.98) | 31±4 (0.96) | 8±2 (0.74) | - (0.44) |
| $R_{SF6}$ (r) | - (0.63) | - (0.48) | 0.19±0.03 (0.91) | 0.17±0.03 (0.94) | 0.66±0.16 (0.76) | - (0.38) |

498
499



**Data availability**

Our $CO_2$, CO, $SF_6$ data from AMY and NWR can be downloaded from ftp://aftp.cmdl.noaa.gov/data/trace_gases. $\Delta(^{14}CO_2)$ data are provided in the supplementary material of this paper.

**Author contributions**

HL wrote this paper and analyzed all data. HL and GWL designed this study. EJD and JCT guided and reviewed this paper. SL collected samples and gave the information of the data at AMY. EJD, JCT, SJL, JBM, GP, and JL provided data and reviewed the manuscript. SSL and YSP reviewed this paper. All authors contributed this work.

**ACKNOWLEDGMENT**

This work was funded by the Korea Meteorological Administration Research and Development Program "Research and Development for KMA Weather, Climate, and Earth system Services– Development of Monitoring and Analysis Techniques for Atmospheric Composition in Korea" under Grant (1365003041).

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
