# Peer review of "Observations of atmospheric $^{14}\text{CO}_2$ at Anmyeondo GAW station,"

_Atmospheric Chemistry and Physics, 2020_

## Referee Comment (RC1) · Anonymous Referee #1 · 5 May 2020

Review of "14C observations of atmospheric CO2 at Anmyeondo GAW station, Korea: Implications for fossil fuel CO2 and emission ratios" by Lee, H. et al. ACPDiss..

General: The manuscript discusses radiocarbon estimated fossil fuel CO2 emissions from local South Korean sources as well as from the Asian continent based on samples taken at the GAW station Anmyeondo in Korea. Additionally, they calculated the emission ratios of CO/CO2 and SF6/CO2 and draw conclusions about improved oxidation efficiency in both the Asian continent as well as Korea. They also state based on a comparison between top-down and bottom-up (inventory) methods that that there is a mismatch of estimated emissions to the point that inventory-based methods lead to up

to 1.8 times lower emissions.

The paper is well written, easy to follow and well-illustrated with graphs. I suggest publications of this manuscript after minor revision:

Minor points: L:434 In South Korea and China, atmosphere-based RCO values are 1.2 times and (1.8$\pm$0.2) times greater than in the inventory, respectively. Please add also an uncertainty for the Korean value.

L: 38 the CO2 increase rate seems very high to me with a large uncertainty, 2.4+-0.5 ppm

L: 56 . . ., since those (not clear what you mean here, I guess CO2)

L:82-83 Why was the station location changed between the previous and the present study?

L:126-127: what about permeation problems associated with glass flasks? To which pressure are the flasks filled? Under which conditions are the flask stored until measurement take place? How long does it take to be analysed?

L:164-166: It might be worthwhile to give a upper limit estimate for this influence. Maybe, also CO2 flux values for the Yellow and Japanese Sea would be helpful for the reader to underpin your conclusion.

Eq. 4-6 I guess these equations are well-known and not necessary to be shown again. I would skip it and only reference on a paper describing this or to the software tool that you have used to calculate the regressions.

L:309-310 This is an important issue to be discussed in more detail, since this relevant with the conclusions drawn from the data about Asian emissions.

L:314 what about correlation between SF6 and CO?

L: 337 what about a contamination from the local SF6 emissions on the ratio assigned

to the Asian continent? Could you get an handle on it from SF6/CO ratios?

Fig. 2 How sensitive are the results on the selection of the background values? To use NWR as background sounds rather strange as the two stations are very far apart and the authors mention explicitly Chinese station as well. Alternatives would be a Japanese location? or a European station. Or even lower bound values of the AMY station based on Hysplit selection.

Table 1 strange that r is low for PL trajectories. Has it to do with only a few values, since there is a much larger addition of fossil fuel CO2 present.

---

## Referee Comment (RC2) · Anonymous Referee #2 · 18 May 2020

This is a well-written paper with interesting data concerning atmospheric observations and validations of fossil CO2, and CO and SF6 emissions from Korea, and the Asian main land. Upon reading, I have made notes and comments that I present below. A more general remark (also given below) is that I invite/encourage the authors to more explicitly conclude what their observations show concerning the quality of the inventories, and if these inventories are thrusworthy or not. By drawing conclusions in that style, these data will be more accessible and valuable to policy makers, and might help to improve the inventories.

I recommend publication after the authors have dealt with my comments below.

[Figure]

Comments ACP-2020-122

page 5, lines 108-109. at what flow rate are the flasks filled, or rather: is the flask air composition an average over some period of time, or merely a point in time?

line 107 "Two pairs of flask-air samples (4 flasks total," In tabel S1 I see only one value per week. Is this an average? Flasks taken together for 14C mm?

line 129 "suggested" ??

line 245, 246 This largest positive Cbio is actually a single point. Which trajectory belongs to this point? The tic marks in the histogram of fig 2C do not correspond to those in the left part of fig 2C.

Lines 252-254. I doubt the explanation offered. Even though your sampling time is early afternoon, I expect still the average mixing height to be the most important player in the mixing ratios of Cff and Cbio, as it influences the flux-to-mixing ratios relation. There must be a seasonal effect in the mixing height no doubt. This must be taken into account in this dicussion. The fact that Cbio covaries with Cff also points to the importance of the (average) mixing height.

Line 270-271 I would say in general nobody expects the CO2 enhancements above background to be entirely due to Cff.

Line 280-282 "During the experimental period, the averages from Asian continent (sectors CE and CN) were higher than KL without the baseline level."

Without the baseline level?? What do you mean?

282 Does OB fit in this set? You call it ocean background, but at the same time you mention it crosses over Shanghai (213-214). In line 234-235 you indicate it again as being background, and then here (282) you take it along with the "real" continental trajectories. This needs to be clarified.

298-299 "we also see CN originated from northeast China and it was around

(10.6±6.9) $\mu$mol mol–1." I don't get the meaning or consequence of this sentence part.

Lines 300-302 Once more, I think average mixing height is the key player here. Are the weather patterns, and thus mixing heights different in the years 2009-2010 from 2014-2106? Did Turnbull et all also sample between 14 and 16 hours?

304 increase of 16.7% line306 "broadly consistent" I disagree for the China case, as you find way larger increases between 2010 and 2016. So you might hypothesize that your measurements indicate much higher increases in fossil fuel use? Of course what you state in 311-312 is very true...

Line 321 are these differences significant? I would say (KL,PL) > (CN,CE) > (CB,OB)

Line 331 To my opinion SF6 is not a good tracer/surrogate for fossil fuel CO2, as it is not produced in the same process. So SF6 actually traces specific industrial activities, and electricity use. Both are coupled to fossil fuel CO2, but not in a 1:1 (spatial,temporal) relation. CO, on the other hand is really co-produced with fossil fuel CO2 (and with biofuel CO2), albeit at a varying rate.

Line 332 "Even though" I don't see the contrast between the strong correlation and the differences

Caption figure S3: "From 2005,.. " -> "From 2005 onwards, .."

lines 347-349 Still, in spite of the still large uncertainty, I invite you to make a stronger statement here, namely that the SF6 inventory in EDGAR and in KNIR are too low given your measurements.

Line 351 also here, watch the significance. I would conclude from table 1 that CB=KL And indeed (see my point higher up), OB is mostly regional background air.

354 "CO...it is more closely related to fossil fuel CO2 emissions" yes, but also to bio-material combustion (compare the Cff to the CO excess)

357-358 I think you can safely erase the word "likely" here. 358 add "and the use of catalysers" ?

358-360 Indeed, biomaterial combustion must play a role, regarding the low Cff especially for CB.

366 Figure S2 -> Figure S1

369 Paragraph 3.4 I suppose you did a similar thing for the SF6 inventories. That means the either the start of this paragraph should be moved up into 3.3, or the SF6 inventory discussions should be taken form 3.3 and moved to this paragraph.

377 "The uncertainty of EDGAR4.3.2 emissions" -> "The uncertainty of EDGAR4.3.2 fossil fuel CO2 emissions"

397-399 if a difference is not significant, it is doubtful to discuss its possible causes.

403 "KNIR seems to have uncounted CO emissions," -> "KNIR suffers from a high number of missing CO emission sources," in other words: make this statement stronger, as the difference is huge: $\approx$2500 vs $\approx$700 Gg in 2012 ! And your data corroborate the Edgar emission ratios...

433-439 S. Korea: your RCO results are 1.2 times the Edgar results. That is hard to see in figure 4. Your value (from table 1) is 8$\pm$2, so a $\pm$25% uncertainty, which makes this factor 1.2 not significant. The Chinese inventories, on the other hand, ARE significantly too low, even though the declining trend has been confirmed by atmospheric measurements. My guess would be that the lack of biofuels/biomaterial burning which is not present in the EDGAR CO inventory, explains the large difference in China, and is not so important in S. Korea.

441 (and also earlier and further) you express mean values $\pm$ standard deviations, whereas the way you write it suggests that this is the error in the mean value, which is in fact sqrt(#mm) lower. So in fact the mean value here is (-6.2$\pm$2.2) ‰ (I took N=70), with a spread of 19 ‰ In your case, most of the time the spread= the standard deviation

is the important feature, but if you compare in lines 446-447 to previous measurements at TAP it is important to know how many measurements those were, and thus what the mean and error in the mean are. Your statement: the average is twice as high strongly suggests that this difference is significant, but the reader can only judge that if you present the error in the mean in both cases. I advise to make this difference between standard deviation and error in the mean clear at the various points where it matters in the paper.

Lines 449-452 Yes, Cff really increased for the air masses from the Asian mainland. Do you conclude that this indicates stronger growth of fossil fuel use than the statistics say? If you think your data clearly point at that, mention that here.

lines 453-463 Based on your data I would (also) conclude the following: (1) 14C analysis is a reliable way of determining Cff in the mixing ratio of air masses (2) Then, the ratio of the emission of rare trace gases and Cff can be determined as well (3) As the inventories for various other trace gases/greenhouse gases are generally much less reliable than that of Cff, these inventories can be validated/verified using atmospheric measurements like ours. (4) I our case we conclude that the inventories for SF6 ... and for CO ...

In this way your results will probably be more valuable to policy makers.

I would also formulate (part of ) this reasoning in the abstract.

Two more references suggested:

Page 2 I would suggest in addition the reference : van der Laan, S. et al. Observation-based estimates of fossil fuel-derived CO2 emissions in the Netherlands using Delta 14C, CO and 222Radon, Tellus B, 62(5, SI), 389–402, doi:10.1111/j.1600-0889.2010.00493.x, 2010.

page 3 line 64 "..correlate well..." I think the earliest 14C-based reference to this is Zondervan, A. and Meijer, H. A. J.: Isotopic characterisation of CO2 sources during

regional pollution events using isotopic and radiocarbon analysis, TELLUS SERIES B-CHEMICAL AND PHYSICAL METEOROLOGY, 48(4), 601–612, doi:10.1034/j.1600-0889.1996.00013.x, 1996.

---

## Author Comment (AC1) · 25 Jul 2020

**Authors' responses to reviewer's comments follow. A copy of the reviewer comment is given (with comment 'number') followed by a response (blue font).**

**Response to referee 1**

1. General comments

   The manuscript discusses radiocarbon estimated fossil fuel $CO_2$ emissions from local South Korean sources as well as from the Asian continent based on samples taken at the GAW station Anmyeondo in Korea. Additionally, they calculated the emission ratios of $CO/CO_2$ and $SF_6/CO_2$ and draw conclusions about improved oxidation efficiency in both the Asian continent as well as Korea. They also state based on a comparison between top-down and bottom-up (inventory) methods that there is a mismatch of estimated emissions to the point that inventory-based methods lead to up to 1.8 times lower emissions. The paper is well written, easy to follow and well-illustrated with graphs. I suggest publications of this manuscript after minor revision:

   We thank you for your comments on the paper's value. We also appreciate your helpful comments to improve our manuscript. According to your specific comments, we revised our manuscript.

2. L:434 In South Korea and China, atmosphere-based RCO values are 1.2 times and (1.8±0.2) times greater than in the inventory, respectively. Please add also an uncertainty for the Korean value.

   Thank you for the comment. We calculated each uncertainty for each sector. And we revised the sentence below.

   **Line 480: In South Korea and China, atmosphere-based $R_{CO}$ values calculated by this study are (1.2±0.3) times (with KL), (1.6±0.4), (1.7±0.4), (2±0.1) and (1.7±0.2) times greater (with CB, CN, CE and OB) than in the inventory, respectively (Figure 4).**

   Also in the abstracts

**Line 33: …originating in China showed (1.6±0.4) to (2±0.1) times greater R$_{CO}$ than…**

In summary as well,

**Line 516: For CO, our values are (1.2±0.3) times and (1.6±0.4) to (2±0.1) times greater …**

3. L: 38 the $CO_2$ increase rate seems very high to me with a large uncertainty, 2.4+-0.5 ppm.

Thank you for the comment. Recently atmospheric $CO_2$ growth rate increased faster than the early measurement period, 1960s (0.8±0.3 ppm/year). ±0.5 is not uncertainty, rather the standard deviation of the annual increases. The value of S.D was a typo and should be ±0.4.

**Line 41: atmosphere at (2.4±0.4) µmol mol$^{-1}$ a$^{-1}$ in a recent decade globally (where 0.4 is the standard deviation of annual growth rates; www.esrl.noaa.gov/gmd/ccgg/trends/, last access: 6 December 2019).**

From 2010 to 2019, the $CO_2$ global annual increase

| Year | 2010 | 2011 | 2012 | 2013 | 2014 | 2015 | 2016 | 2017 | 2018 | 2019 | Mean (±S.D.) |
|---|---|---|---|---|---|---|---|---|---|---|---|
| ppm/year | 2.4 | 1.7 | 2.4 | 2.4 | 2.0 | 3.0 | 2.9 | 2.1 | 2.4 | 2.6 | 2.4±0.4 |

These values are from www.esrl.noaa.gov/gmd/ccgg/trends/, as we citied in the manuscript.

4. L: 56 . . ., since those (not clear what you mean here, I guess $CO_2$)

Corrected.

**Line 60: We revised it from 'those emissions' to 'fossil fuel $CO_2$ emissions'**

5. L:82-83 Why was the station location changed between the previous and the present study?

The TAP station does not belong to the KMA/NIMS and this paper focuses only on data from AMY. Therefore we think our data can give the information of this region with recent data since AMY is close to TAP (28 km away from AMY). We did not add that specific information in the manuscript.

6. L:126-127: what about permeation problems associated with glass flasks? To which pressure are the flasks filled? Under which conditions are the flask stored until measurement take place? How long does it take to be analysed?

The flasks have undergone extensive laboratory testing to ensure they maintain sample integrity for storage times up to one year. Comparison of flask-air samples with in situ measurements at South Pole have revealed storage offsets of up to 0.2 ppm after a year, but storage times at AMY are much less, and the difference in pressure between the flask and outside air (the main driver of preferential diffusion through the Teflon o-rings) is also less.

Flask-air sampling steps are as follows;

Using a semi-automated sampler, flasks are flushed at 5-6 L/min for 10 min then pressurized to 5 – 6 psig. After the pump turns off, falling pressure indicates a leak at the connectors. In that case, flasks are reconnected and the sample collected again. To prevent a slow leak through the pump, we close the stopcocks from the pump and to the exhaust first (esrl.noaa.gov/gmd/ccgg/psu/mannuals/psu_manual_1.6.pdf). We store the collected samples in the laboratory and send them to NOAA about every two months.

Another reason we are sure there is no permeation problem is that we compare the flask-air $CO_2$ data to KMA continuous measurements. We confirmed the differences are small close to GAW's compatibility goal (±0.1 ppm; Lee et al., 2019).

We added the sentence in the manuscript

**Line 114: Two pairs of flask-air samples (4 flasks total, 2 L, borosilicate glass with Teflon O-ring sealed stopcocks) were collected about weekly from a 40 m tall tower at AMY, regardless of wind direction and speed from May 2014 to August 2016, generally between 1400 to 1600 local time (Table S1) using a semi-automated portable sampler. A pair of flasks was flushed for 10 min at 5-6 L min$^{-1}$ then pressurized to 5.5 psig in less than 1 min. A second pair is collected shortly after the first (within 20 min). The portable sampler was checked for leaks after pressurizing by observing the pressure gauge before closing the stopcocks. Batches of sampled flasks were shipped to Boulder, CO, USA every two months.**

**Line 136: When we compare NOAA's $CO_2$ measurements from flask-air with quasi-continuous measurements by KMA at AMY, the difference was -0.11±2.32 µmol mol-1 (mean±1 σ), close to GAW's compatibility goal for $CO_2$ (±0.1 ppm for Northern Hemisphere measurements, Lee et al., 2019).**

Reference: www. esrl.noaa.gov/gmd/ccgg/flask.html

7.  L:164-166: It might be worthwhile to give a upper limit estimate for this influence. Maybe, also $CO_2$ flux values for the Yellow and Japanese Sea would be helpful for the reader to underpin your conclusion.

We agree with the reviewer It would be great to test whether the samples were affected by ocean fluxes, but this is well-beyond this study. So we added more references. There is a reference value of a flux that estimate for the East China Sea of -4.2 mmol/m$^2$/day (Song et al., 2018). This value is very negligible. Turnbull et al. (2009) reported no significant bias from oceans in the Northern Hemisphere, even at coastal sites, while this bias is very important in the Southern Hemisphere. Also Turnbull et al. (2011a) mentioned that ocean exchange was negligible at TAP. Therefore we just added this reference in the manuscript and explained the bias from the ocean can be negligible.

**Line 184: It was also demonstrated there is no significant bias from the oceans including**

**East China Sea (Song et al., 2018), even at coastal sites in Northern Hemisphere (Turnbull et al., 2009).**

8. Eq. 4-6 I guess these equations are well-known and not necessary to be shown again. I would skip it and only reference on a paper describing this or to the software tool that you have used to calculate the regressions.

We deleted and added the reference. On the other hand, to make readers understand easily, we described the equations in the supplementary materials.

**Line 201: we use reduced major axis (RMA) regression analysis (Sokal and Rohlf, 1981)**

**Line 205: The relevant equations are presented from Equ. S1 to Equ. S3.**

**Delete the equations from Line 207 to 217**

9. L:309-310 This is an important issue to be discussed in more detail, since this relevant with the conclusions drawn from the data about Asian emissions.

It is very clear that even the $C_{ff}$ from CB sector in this study increased compared to TAP far-field samples from 2004/2010. CB sector is the cleanest sector in this study with high wind speed (median value is 5 m/s and maximum of 10.2 m/s) and high PBL (median value is 600 m and maximized up to 1700 m). Therefore we assumed that any contamination could not affect the samples due to synoptic condition. For other sector which are originated from China, not only this study but also other studies showed the increased values compared to the Turnbull et al.(2011a).

But we did not totally ignore the possibility. As reviewer mentioned, it would be great to mention about those factors in more detail and to consider for further study.

**Line 347: On the other hand, those values from this study showed large variability with small sample numbers due to different sampling strategy, environment, and synoptic**

**conditions such as boundary layer height at the sampling time from reference studies. Further study will be necessary to understand those increased values.**

10. L:314 what about correlation between $SF_6$ and CO?

When we implement RMA analysis for $CO/SF_6$, the correlation is very weak (R=0.18). And to consider only the outflow of Asian continent, R was 0.24. Only CE and OB whose CO and $SF_6$ had a good correlation with $C_{ff}$ showed good correlation (R>0.6).

11. L: 337 what about a contamination from the local SF6 emissions on the ratio assigned to the Asian continent? Could you get an handle on it from SF6/CO ratios?

We considered this idea, when analyzing the Rgas values. But it was not possible due to several reasons. 1) as we explained, basically the correlation between $SF_6$ and CO was weak. 2) To select the data with $SF_6/CO$ ratio, the ratios of sample-by-sample should be constant (or Gaussian). However the data characteristics did not show that.

Therefore to reduce local $SF_6$ effects, after cluster analysis we select the data again for wind speed > 3 m/s, as described in section 3.2. As seen in Table 1, the mean value and standard deviation of $SF_6$ in outflow from the Asia continent is smaller than for South Korea. This also means that $SF_6$ values was not be affected by local effects as shown by relatively constant values. We have high confidence that ratios from the Asian continent are less affected by local pollution.

12. Fig. 2 How sensitive are the results on the selection of the background values? To use NWR as background sounds rather strange as the two stations are very far apart and the authors mention explicitly Chinese station as well. Alternatives would be a Japanese location? or a European station. Or even lower bound values of the AMY station based on Hysplit selection.

When we selected the baseline station, there were only a few possible stations where $^{14}C$ in $CO_2$ data were available. Asian stations would be a good option for this study but there is no available $^{14}C$ data. And even if a data set existed, when the sampling/analysis methods are different, the

data uncertainty can be increased. Therefore we used NWR data, which are located at similar latitude to AMY with the same sampling/analysis method used as at AMY. And the analysis for $^{14}$C was conducted by the same institute, INSTAAR, thus decreasing uncertainty that might occur if measurements from different laboratories were used (Miller et al., 2013)

According to Turnbull et al. (2011a), choice of background values did not significantly influence derived enhancements due to the large regional and local signal at TAP, 28 km from AMY. It was also described on Line 160. We hope the reviewer can understand that $^{14}$C data are limited, and this is the one of reasons which makes this paper valuable.

Reference: Miller et al., (2013), Initial results of an intercomparison of AMS-Based atmospheric $^{14}CO_2$ measurements, Radiocarbon, Vol.55, Nr 2-3, 2013, 1475-1483.

13. Table 1 strange that r is low for PL trajectories. Has it to do with only a few values, since there is a much larger addition of fossil fuel $CO_2$ present.

When sampling well-mixed air masses, we can clearly see correlations among the gases. Under stagnant conditions, due to micro scale meteorology, the gases showed correlated weakly.

---

## Author Comment (AC2) · 25 Jul 2020

Authors' responses to reviewer's comments follow. A copy of the reviewer comment is given (with comment 'number') followed by a response (blue font).

**Response to referee 2**

1. General comments

This is a well-written paper with interesting data concerning atmospheric observations and validations of fossil CO2, and CO and SF6 emissions from Korea, and the Asian main land. Upon reading, I have made notes and comments that I present below. A more general remark (also given below) is that I invite/encourage the authors to more explicitly conclude what their observations show concerning the quality of the inventories, and if these inventories are thrus worthy or not. By drawing conclusions in that style, these data will be more accessible and valuable to policy makers, and might help to improve the inventories. I recommend publication after the authors have dealt with my comments below.

We thank you for your comments on this paper's value. We also appreciate your helpful comments to improve our manuscript. According to your specific comments, we revised our manuscript.

2. page 5, lines 108-109. at what flow rate are the flasks filled, or rather: is the flask air composition an average over some period of time, or merely a point in time?.

Samples are collected with a semi-automated sampler that first flushes the flasks at 5-6 L/min for 10 minutes then pressurizes them to 5.5 psig (5-6 LPM) in less than 1 min. Therefore, samples are integrated over 1-2 min.

We added the sentence on line 114.

Line 114: Two pairs of flask-air samples (4 flasks total, 2 L, borosilicate glass with Teflon O-ring sealed stopcocks) were collected about weekly from a 40 m tall tower at AMY, regardless of wind direction and speed from May 2014 to August 2016, generally between 1400 to 1600 local time (Table

S1) using a semi-automated portable sampler. A pair of flasks was flushed for 10 min at 5-6 L min-1 then pressurized to 5.5 psig in less than 1 min. A second pair is collected shortly after the first (within 20 min). The portable sampler was checked for leaks after pressurizing by observing the pressure gauge before closing the stopcocks, Batches of sampled flasks were shipped to Boulder, CO, USA every two months

line 107 "Two pairs of flask-air samples (4 flasks total," In tabel S1 I see only one value per week.
 Is this an average? Flasks taken together for 14C mm?

We collect 4 samples but 2 among them were analyzed for 14C in CO2. Among four flasks, the air from two flasks, after analysis for greenhouse gas mole fractions, was combined and analyzed for  $\Delta$ (14CO2)

Therefore we added the relevant sentence on Line131

Line144: Among four flasks, the air from two flasks, after analysis for greenhouse gas mole fractions, was combined and analyzed for  $\Delta$ (14CO2)

4. line 129 "suggested" ??

Line 144 we revised it to" tabulated"

5. line 245, 246 This largest positive  $C_{bio}$  is actually a single point. Which trajectory belongs to this point? The tic marks in the histogram of fig 2C do not correspond to those in the left part of fig 2C.

The largest positive value of  $C_{\text{bio}}$  is observed on August 2 2016 as shown in Figure 2. And it was observed in PL sector. To make it clear we revised the sentence line 279.

Line 287: The highest Cbio value was also observed in summer, PL sector.

Thank you for pointing out the y-axis ticks. We revised Figure 2 as below.

Figure 2. Time series of (a) observed CO2 dry air mole fraction (open circles) and observed CO2 (Cobs) minus Cff calculated from  $\Delta$ (14CO2) (closed circles). (b)  $\Delta$ (14CO2) at AMY (black circles) and at NWR (Niwot Ridge, line), baseline data. (c) Time series of Cff and Cbio calculated from  $\Delta$ (14CO2) (left) and the frequency distribution at AMY (right).

6. Lines 252-254. I doubt the explanation offered. Even though your sampling time is early afternoon, I expect still the average mixing height to be the most important player in the mixing ratios of Cff and Cbio, as it influences the flux-to-mixing ratios relation. There must be a seasonal effect in the mixing height no doubt. This must be taken into account in this discussion. The fact that Cbio covaries with Cff also points to the importance of the (average) mixing height.

We agreed with reviewer's comments. When we analyzed the PBL height for each sample by meteorological model, winter was highest (with a range from 150 m to 1100 m) and summer/spring are lower than other seasons (with a range from 100 m to 800 m). This result is consistent with the explanation of wind speed in the manuscript. Therefore we added this sentence.

Line 282: When we analyzed seasonal boundary layer height for each sample by UM-GDAPS, it also showed similar result that it was highest in winter (with a range from 150 m to 1100 m) and lowest in summer (with a range from 100 m to 500 m). This suggests that these high summer Cff values may reflect emission from local activities, which were described in section 2.1, more than in other seasons.

 Line 270-271 I would say in general nobody expects the CO2 enhancements above background to be entirely due to Cff.

Thank you for the comment. The point is that regardless of the source and seasons, we find that  $C_{bio}$  contributes to atmospheric  $CO_2$  enhancements at AMY. And if we just use  $CO_2$  enhancements to demonstrate the bottom-up inventory, it can be biased. Therefore we emphasized this in section 4. "Finally, we stress that because  $C_{bio}$  contributes substantially to  $\Delta x(CO_2)$ , even in winter, 14C-based  $C_{ff}$  (and not  $\Delta x(CO_2)$ ) is required for accurate calculation of both  $R_{co}$  and  $R_{SF6}$ ."

To avoid the confusion, we revised the explanation in the section directly on line 302 Line 302: ... so when only  $CO_2$  enhancements above background are compared to bottomup inventories, it can make a bias due to  $C_{bio}$  contributions. 8. Line 280-282 "During the experimental period, the averages from Asian continent (sectors CE and CN) were higher than KL without the baseline level." Without the baseline level?? What do you mean?

"Without baseline" means that Continental Baseline (CB) and Ocean Baseline (OB) sector.

We added the explanation on line 316,

**Line 316: ...without baseline sector (CB and OB)**

9. Does OB fit in this set? You call it ocean background, but at the same time you mention it crosses over Shanghai (213-214). In line 234-235 you indicate it again as being background, and then here (282) you take it along with the "real" continental trajectories. This needs to be clarified.

We agree. The cluster definitely belongs to Ocean Background, but some of the air masses (4 of 10) have paths over southern China, such as Shanghai. We confirmed that the altitudes that air masses came through were high enough when over southern China that they were unlikely affected by surface sources and sinks. But we cannot ignore the possibility that these trajectories could have affected our results. We revised the sentence below:

Line 239: .Among them, a few of the trajectories passed over the eastern part of China (e.g., over Shanghai) with high altitude (~1000 m).

10. 298-299 "we also see CN originated from northeast China and it was around (10.6±6.9) μmol mol–
1." I don't get the meaning or consequence of this sentence part.

A previous study mentioned that northeast China affected SDZ-NE sector samples and they showed  $(3\pm7) \mu$ mol mol-1 in 2009 to 2010 and increased to  $(7.6\pm6.8) \mu$ mol mol–1 in 2015.

In this study, samples originated from northeast China (NE) showed increased  $C_{\rm ff}$  levels around (10.6±6.9) µmol compared to 2009 to 2010.

We revised the sentence clear

Line 333: we also see CN that originated from northeast china (NE) and its mean value of  $C_{ff}$  had increased around (10.6±6.9) µmol mol-1 compared to those values in 2009 to 2010.

11. Lines 300-302 Once more, I think average mixing height is the key player here. Are the weather patterns, and thus mixing heights different in the years 2009-2010 from 2014-2106? Did Turnbull et all also sample between 14 and 16 hours?

Turnbull et al. (2011a) sampled flasks during mid-afternoon, so that we assume the sampling time might be similar. One difference is they collected air when there was an onshore wind while in this study, samples were collected regardless of wind direction and speed.

We could not research the mixing height in 2009 to 2010, but it is very clear that  $C_{\rm ff}$  from CB sector in this study increased compared to TAP far-field samples from 2004/2010. CB sector is the cleanest sector in this study with high wind speed (median value is 5 m/s, with a maximum of 10.2 m/s) and high PBL (median value is 600 m, with a maximum of 1700 m). There is no possibility for local-scale pollution to affect the samples due to the synoptic conditions.

Other sectors that originated from China, not only in this study, but also other studies, showed increased values compared to Turnbull et al. (2011a).

On the other hand, we also considered the possibility of unexpected conditions that could affect this analysis. *Line 345: It is also likely that the proximity of local emission sources to AMY is causing higher observed C*ff under some synoptic conditions.

As the reviewer mentioned, it would be great to mention those factors in more detail and the importance for further study, so we removed the sentence above and added:

Line 347: On the other hand, those values from this study showed large variability with small sample numbers due to different sampling strategy, environment, and synoptic conditions such as boundary layer height at the sampling time from reference studies. Further study will be necessary to understand those increased values.

12. 304 increase of 16.7% line306 "broadly consistent" I disagree for the China case, as you find way larger increases between 2010 and 2016. So you might hypothesize that your measurements indicate much higher increases in fossil fuel use? Of course what you state in 311-312 is very true...

We also pointed out the Asia main land differently from Korea Local air.

Line 341: This is broadly consistent with the flat trend in observed  $C_{\rm ff}$  in KL air masses, and in the upward trend in  $C_{\rm ff}$  observed in air-masses flowing out from Asia.

We also explained the possibilities to affect this result on line 347. Therefore we did not revise the sentence.

13. Line 321 are these differences significant? I would say (KL,PL) > (CN,CE) > (CB,OB)

**Line358: Corrected**

14. Line 331 To my opinion SF6 is not a good tracer/surrogate for fossil fuel CO2, as it is not produced in the same process. So SF6 actually traces specific industrial activities, and electricity use. Both are coupled to fossil fuel CO2, but not in a 1:1 (spatial, temporal) relation. CO, on the other hand is really co-produced with fossil fuel CO2 (and with biofuel CO2), albeit at a varying rate.

Though the reviewer's comment is true, it is also true that  $C_{ff}$  and  $\Delta SF_6$  correlate quite strongly in CE, OB and KL sectors. This means even if they don't have a same source, they are emitted from similar regions. Therefore, we suggest that  $\Delta SF_6$  can be a proxy of fossil fuel CO2 in those regions. But as you mentioned  $\Delta SF_6$  cannot always a good tracer of fossil fuel CO2, so we revised the sentence as below. We revised line 362:

Line 372: Thus SF6 can be a good tracer of fossil fuel  $CO_2$  for those regions.

15. Line 332 "Even though" I don't see the contrast between the strong correlation and the differences

The correlation was strong in both South Korea and for the Asian continent, but the R value was totally different for both regions.

We revised line 373:

Line 373: The correlation between  $\Delta x$ (SF6) and Cff was strong in CE, OB and KL, however, RSF6 is different between South Korea and outflow from the Asian continent (Figure S2).

16. Caption figure S3: "From 2005,.. " -> "From 2005 onwards, .."

**Corrected.**

17. lines 347-349 Still, in spite of the still large uncertainty, I invite you to make a stronger statement here, namely that the  $SF_6$  inventory in EDGAR and in KNIR are too low given your measurements.

**We revised the sentence here**

Line 388: Even though KL  $R_{SF6}$  showed greater uncertainty than CE and OB, it is still greater than bottom-up inventories, such as KNIR and EDGAR. Therefore it would be useful to get more data to try and derive a more robust estimate to evaluate  $SF_6$  emission inventories for Korea.

18. Line 351 also here, watch the significance. I would conclude from table 1 that CB=KL. And indeed

(see my point higher up), OB is mostly regional background air.

Corrected (Line 395).

For OB sector, we revised the explanation according to reviewer's comment No.9 above.

19. 354 "CO...it is more closely related to fossil fuel CO2 emissions" yes, but also to biomaterial combustion (compare the Cff to the CO excess)

We agree. The revised sentence below

Line 397: Since CO is produced during incomplete combustion of fossil fuels and biomass, it is more closely related to fossil fuel  $CO_2$  emissions than the other trace gases.

20. 357-358 I think you can safely erase the word "likely" here. 358 add "and the use of catalysers" ?

**Corrected**

Line 402 ... due to differences in combustion efficiencies and the use of catalytic converters

21. 358-360 Indeed, biomaterial combustion must play a role, regarding the low Cff especially for CB.

We agree and added the following sentence.

Line 405: For example, for CB the CO level is similar to KL while  $R_{co}$  is higher than KL with low Cff.

22. 366 Figure S2 -> Figure S1

Corrected (Line 413).

23. 369 Paragraph 3.4 I suppose you did a similar thing for the SF6 inventories. That means the either the start of this paragraph should be moved up into 3.3, or the SF6 inventory discussions should be taken form 3.3 and moved to this paragraph.

Corrected. We separated two sections according to the species.

- 3.3 Correlation of Cff with SF6 and its emission ratios
- 3.4 Correlation of Cff with CO and its emission ratios
- 24. 377 "The uncertainty of EDGAR4.3.2 emissions" -> "The uncertainty of EDGAR4.3.2 fossil fuel CO2 emissions"

**Corrected (Line 422)**

25. 397-399 if a difference is not significant, it is doubtful to discuss its possible causes.

**We removed the sentence from Line 443 to Line 444.**

26. 403 "KNIR seems to have uncounted CO emissions," -> "KNIR suffers from a high number of missing CO emission sources," in other words: make this statement stronger, as the difference is huge: \_2500 vs \_700 Gg in 2012 ! And your data corroborate the Edgar emission ratios...

**Corrected**

Line 448: KNIR suffers from a large number of missing CO emission sources compared to the EDGAR, as indicated by their reported emissions, 638.3 and 2580.8 Gg in 2012, respectively

27. 433-439 S. Korea: your RCO results are 1.2 times the Edgar results. That is hard to see in figure 4.

Your value (from table 1) is 8±2, so a ±25% uncertainty, which makes this factor 1.2 not significant. The Chinese inventories, on the other hand, ARE significantly too low, even though the declining trend has been confirmed by atmospheric measurements. My guess would be that the lack of biofuels/biomaterial burning which is not present in the EDGAR CO inventory, explains the large difference in China, and is not so important in S. Korea.

We agree reviewer's comment for two reasons. Frist, EDGAR does not reflect secondary CO production and, second, CO derived from biomass burning and biofuels was not included by EDGAR. Since we described it already, we did not revise the sentence. This is described in the manuscript as:

Line 485: Also, CO derived from biomass burning and biofuels was not included in this inventory. Therefore, this indicates that top-down observations are necessary to evaluate and improve bottom-up emission products.

28. 441 (and also earlier and further) you express mean values ± standard deviations, whereas the way you write it suggests that this is the error in the mean value, which is in fact sqrt(#mm) lower. So in fact the mean value here is (-6.2±2.2) ‰ (I took N=70), with a spread of 19‰. In your case, most of the time the spread= the standard deviation is the important feature, but if you compare in lines 446-447 to previous measurements at TAP it is important to know how many measurements those were, and thus what the mean and error in the mean are. Your statement: the average is twice as high strongly suggests that this difference is significant, but the reader can only judge that if you present the error in the mean in both cases. I advise to make this difference between standard deviation and error in the mean clear at the various points where it matters in the paper.

Thank you for your comment. It is very true that standard deviation (SD) and standard error of the mean (SEM) are totally different. The reference values from Turnbull et al. (2011a) and Niu et al. (2016) were also SD rather than SEM, which we verified in their publications. As the reviewer already commented, SD is the dispersion of data in normal distributions. In other words, SD indicates how accurately the mean represents sample data. For SEM, it is the SD of the theoretical distribution of the sample mean (the sampling distribution).

In this regards, it would be good to use SEM though, however, since the data set we used here is not continuous, only weekly resolution, the number of data is quite small for SEM. When we use SEM, the data characteristics can be underestimated because the error can be decreased, certainly. The number of data is very limited that dispersion of data can be important information for reader.

Therefore we just added the number of the data for the calculation (since whether we use SD or SEM, this is very important information) and retain SD.

Please see the revised manuscript. When previous studies did not include the number of data, we could not include it.

29. Lines 449-452 Yes, Cff really increased for the air masses from the Asian mainland. Do you conclude that this indicates stronger growth of fossil fuel use than the statistics say? If you think your data clearly point at that, mention that here.

The atmospheric Cff was increased compared to the previous studies but we cannot explain that our results are much greater than the reported inventory values. When we analyze inverse modeling, we can point it clearly. Therefore we revised sentence in section 4, 2).

Line 502: After separately identifying samples originating from the Asian continent and the Korean peninsula, we determined that the mean  $C_{\rm ff}$  increased relative to the earlier observations due to increased fossil fuel emissions from the Asian continent as showing the consistent growth with reported emission increased 16.7% in China while 1.8% in South Korea from 2010 to 2016.

30. lines 453-463 Based on your data I would (also) conclude the following: (1) 14C analysis is a reliable way of determining Cff in the mixing ratio of air masses (2) Then, the ratio of the emission of rare trace gases and Cff can be determined as well (3) As the inventories for various other trace gases/greenhouse gases are generally much less reliable than that of Cff, these inventories can

be validated/verified using atmospheric measurements like ours. (4) I our case we conclude that the inventories for SF6 ... and for CO ... In this way your results will probably be more valuable to policy makers. would also formulate (part of ) this reasoning in the abstract.

We re-write the summary and conclusion according to reviewer's comment. Please see the revised the version of section 4 and abstract as well.

Two more references suggested: Page 2 I would suggest in addition the reference : van der Laan,
 S. et al. Observation based estimates of fossil fuel-derived CO2 emissions in the Netherlands using Delta 14C, CO and 222Radon, Tellus B, 62(5, SI), 389–402, doi:10.1111/j.1600-0889.2010.00493.x, 2010.

**We added the reference.**

32. page 3 line 64 "...correlate well..." I think the earliest 14C-based reference to this is Zondervan, A. and Meijer, H. A. J.: Isotopic characterisation of CO2 sources during regional pollution events using isotopic and radiocarbon analysis, TELLUS SERIES BCHEMICAL AND PHYSICAL METEOROLOGY, 48(4), 601–612, doi:10.1034/j.1600-0889.1996.00013.x, 1996.

We added the reference.

---

## Author Response (AR2)

**Authors' responses to reviews follow. A copy of the editor's comment is given (with comment 'number') followed by a response (blue font).**

**Response to referee 1**

1. General comments

   The unit of emissions is mass per time. In the present case, emissions are reported over annual periods, so the units should be corrected to "Gg a-1" etc. There are numerous instances in main text.

   We thank you for accepting our paper. We tried to reflect your comment on our manuscript and revised the unit of emission in the main text.

2. l. 25: "a measurement uncertainty"

   L25: Corrected

3. l. 28: "above background mole fractions"

   L28: Corrected

4. l. 32: "2.0±0.1"

   L32: Corrected

   .

5. l. 114: Please convert 5.5 psig to SI units (bar or Pa).

   L114: Corrected to 0.38 bar

6. l. 152: "is the Δ(14CO2) value of ..."

   L152: Corrected: where $\Delta$ is the $\Delta(^{14}C)$ of each $CO_2$ component of Equ. (1)

7. l. 158: Δ(14CO2).

   L158: Corrected: $\Delta(^{14}CO_2)$ is reported as a per mil

8.  l. 159: "≈ R_sample(14C/C) / R_standard(14C/C) - 1, were R(14C/C) is the 14C/C amount ratio" [a chemical symbol alone cannot represent a quantity]

L159: Corrected: $\Delta(^{14}C)$ ≈[R_ sample($^{14}$C/C)/R_ standard($^{14}$C/C)-1]1000‰, where R_($^{14}$C/C) is the $^{14}$C/C amount ratio

9.  l. 169: $\Delta(14CO_2)$

L170: Corrected:… $CO_2$ that have a $\Delta(^{14}C)$ differing by a small

10. l. 223: Please delete the tilde (~) sign. All quantities should be reported with an appropriate number of significant figures and rounded according to their uncertainty.

L224: Corrected: 1000 m

11. l. 492: Please change "absence" to "contribution", for clarity.station. Or even lower bound values of the AMY station based on Hysplit selection.

L493: Corrected: …no contribution of ….

This described that the bottom of inventory missed the sources from oxidation of $CH_4$ and non-methane VOC.

12. Fig. 2a: The y-axis label should be x(CO2)

Fig. 2a indicated atmospheric $CO_2$ level (not a enhancement value) so that we thought just $CO_2$ is right. Therefore we just retained this.

13. Fig. 2b: The y-axis label should be $\Delta(14CO_2)$

Corrected.

[Figure]

14. Table 1: Add "mole fraction" after CO and SF6 in the table caption. The row headings should be

C_ff / (μmol mol-1), x(CO) / (nmol mol–1), x(SF6) / (pmol mol-1), R_CO / (nmol μmol-1) and R_SF6

/ (pmol μmol-1). The units should be removed from the caption. You may want to insert a separate

row 3 to indicate the number of data N and remove this information from row 2. Please add to the

caption the meaning of the abbreviations CB, CN, CE, OB, KL and PL (in case the table is

displayed out of context)

We just retained the description of unit in the caption since the room is too small to include all. We

added the meaning of the abbreviation of CB, CN, CE, OB, KL and PL.

15. Fig. S3: Please change y-axis label to "SF6 emissions (Gg a-1)"

Corrected

16. Fig. S4: Please change y-axis labels to "fossil CO2 emissions (10^5 Gg a-1)" and "CO emissions (10^3 Gg a-1)"

Corrected

17. Fig. S4: Please change y-axis labels to "fossil CO2 emissions (10^4 Gg a-1)" and "CO emissions (10 Gg a-1)"

Corrected